# Long-wave Absorption of Few-Hole Gas in Prolate Ellipsoidal Ge/Si Quantum Dot: Implementation of Analytically Solvable Moshinsky Model

**DOI:** 10.3390/nano10101896

**Published:** 2020-09-23

**Authors:** David B. Hayrapetyan, Eduard M. Kazaryan, Mher A. Mkrtchyan, Hayk A. Sarkisyan

**Affiliations:** 1Institute of Engineering and Physics, Russian-Armenian University, 123 Hovsep Emin Str., Yerevan 0051, Armenia; david.hayrapetyan@rau.am (D.B.H.); edghaz@mail.ru (E.M.K.); mher.mkrtchyan@rau.am (M.A.M.); 2Institute of Physics, Nanotechnology and Telecommunications, Peter the Great St. Petersburg Polytechnic University, 195251 Saint Petersburg, Russia

**Keywords:** Kohn theorem, ellipsoidal quantum dot, hole gas, Moshinsky model, adiabatic approximation

## Abstract

In this paper, the behavior of a heavy hole gas in a strongly prolate ellipsoidal Ge/Si quantum dot has been investigated. Due to the specific geometry of the quantum dot, the interaction between holes is considered one-dimensional. Based on the adiabatic approximation, it is shown that in the z-direction, hole gas is localized in a one-dimensional parabolic well. By modeling the potential of pair interaction between holes in the framework of oscillatory law, the problem is reduced to a one-dimensional, analytically solvable Moshinsky model. The exact energy spectrum of the few-hole gas has been calculated. A detailed analysis of the energy spectrum is presented. The character of long-wave transitions between the center-of-mass levels of the system has been obtained when Kohn theorem is realized.

## 1. Introduction

The investigation of few-particle states in quantum dots (QDs) has academic and applied significances [1,2,3]. QDs are unique objects on the basis of which many fundamental principles of quantum physics of few-particle systems can be tested, and on the other hand, these promising structures can play the role of the functional base for the semiconductor devices of a new generation [3]. Being atom-like systems, QDs can be described using various exact and approximate quantum mechanics methods. However, these systems have advantage over real atoms because a strong dependence of the energy spectrum on the geometrical shape and sizes of QDs allows manipulating the energy spectrum and consequently, the physical characteristics of QDs. So QDs are also called “artificial atoms” [4]. Both one-particle and few-particle systems in quantum dots of various geometric shapes have been theoretically studied (see for example [5,6,7,8,9,10]). For simple geometries (spherical, cylindrical) of QDs, in some cases, the single-particle Schrödinger equations have exact analytical solutions [11,12,13,14,15]. This allows to give a full description of many physical properties of QDs: absorption coefficient, threshold absorption frequencies, orbital single-electron current, etc. [14,15,16].

The problem of describing Coulomb (impurity, exciton) complexes in QDs is already becoming much more complicated. In the case of a spherical QD with infinitely rectangular walls, when the impurity center is localized in the center of the QD, the problem can be solved analytically [17]. However, even a small displacement of the impurity from the center of the QD leads to mathematical difficulties and the problem has to already be solved within the framework of the variational method or in the approximation of the perturbation theory [18,19,20].

The problem of the behavior of a two-electron system in QDs is similar to the problem of a helium atom. We can introduce ortho- and para- states, determine the exchange energy in the framework of the Russell-Saunders approximation, and enter the time of exchange of states between electrons, etc. Based on the foregoing, it is reasonable to use the Hartree-Fock or Thomas-Fermi approximation [21,22,23,24,25]. In paper [24], the many-particle states in a 2D parabolic QD, based on the Thomas-Fermi approximation, were investigated. A remarkable property of such a system was its exact solvability. The authors have shown that the two-dimensional Thomas-Fermi equation reduces to the Bessel equation.

Problems connected with the behavior of few- and many- particle systems in QDs with a parabolic confining potential attracted the attention of specialists after the generalization of Kohn theorem [4,26,27] to the case of parabolic QDs. This theorem was originally formulated for the case of an electron gas in axial magnetic field by Walter Kohn in 1961. Kohn showed that, when in dipole approximation, a long-wavelength electromagnetic perturbation fell on the short-range, pair-interacting electron gas with the interparticle interaction energy with the following form $\sum_{i<j}\upsilon \left(\left|{\vec{r}}_{i}-{\vec{r}}_{j}\right|\right)$, and the frequency of the cyclotron resonance was not affected by the number of electrons. Maxim and Chakraborty generalized this theorem for the case of QDs [4]. In this paper, authors have considered a pair-interacting electron gas localized in two-dimensional symmetric parabolic QDs, in the presence of an axial magnetic field. It was shown that center of mass and relative motions of electrons are separated due to the parabolic form of the confining potential, and as a result, the frequency of resonant absorption of long-wave radiation does not depend on the number of electrons.

The parabolic confining potential can occur due to the effect of diffusion between the QD components and the environment [28]. On the other hand, as was shown in [29], the parabolic confining potential occurs in QDs with a specific geometry, in particular in strongly prolate or oblate ellipsoidal QDs [30]. This fact can be justified on the basis of the adiabatic approximation, when the Hamiltonian of the system can be represented as the sum of two independent Hamiltonians, one of which describes a fast subsystem and the other – a slow one. It can be assumed that in such systems, the generalized Kohn theorem can be implemented. This problem for the case of an electron gas localized in strongly oblate or prolate ellipsoidal QDs has been discussed [31,32,33,34]. Alternatively, in papers [35,36], results on absorption from a hole gas localized in a strongly flattened Ge/Si lens-shaped QD have been presented. Theoretically defined values of resonant frequencies gave a good match with the experimental result.

Kohn’s theorem is realized when long-wave transitions take place between the levels of the center of mass of a many-particle gas. Such levels correspond to single-particle states in a parabolic well. At the same time, it is of interest to determine the spectrum of the relative motion of particles. This problem has been studied in detail in [4,27]. Alternatively, there are examples of exactly solvable many-particle systems localized in a parabolic well with pair-interaction between particles. One of these models is the Moshinsky atom [37]. Initially, this model was proposed to describe the behavior of nucleons in nuclei. Considering a system of pair-interacting nucleons in a confined region of the nucleus, Moshinsky assumed that the particles interact with each other according to the harmonic law, and the system is located in a parabolic well. Later, Johnson and Payne showed that in the case of a two-dimensional parabolic QD, the presence of an oscillator pair-interaction between electrons makes it possible to exactly diagonalize the N-particle Hamiltonian in the presence of an axial magnetic field [38].

As stated above, QDs with the strongly oblate or prolate ellipsoidal or lens-shaped geometries allow the use of adiabatic approximation. Moreover, the confining potential of the slow subsystem is parabolic. If we assume that the QD has the geometry of a strongly prolate ellipsoid, then in the direction of the axis of rotation (z axis), the pair-interacting gas will be localized in a parabolic well. If we take into account that in the radial direction, the interparticle interaction is much weaker than size quantization, then we can neglect it, and in the z-direction, we come to the problem of a one-dimensional, pair-interacting gas localized in a parabolic well. This problem, without specifying interaction potential, has been discussed in [39], in particular, it was shown that the generalized Kohn theorem is realized in such a system. Alternatively, if we adapt the Moshinsky atom model to the description of one-dimensional localized gas, we can obtain an analytical expression for the energy spectrum of the system.

The goal of this paper is analytical investigation of pair-interacting hole gas in a strongly prolate ellipsoidal QD, and demonstration of the realization of generalized Kohn’s theorem in such a structure.

## 2. Adiabatic Description of the Hole Gas

Let us consider pair-interacting, few-hole gas localized in the strongly prolate ellipsoidal QD (Figure 1) with interparticle interaction potential of the following type.
(1)$$V_{\mathrm{int}}\left(1,\ldots ,N\right)=\sum_{i<j}\upsilon \left(\left|{\vec{r}}_{i}-{\vec{r}}_{j}\right|\right)$$

By the analogy with [33,34], the confining potential of QD for the relatively low levels we choose is in the following form.
(2)$${\hat{V}}_{conf}\left(\vec{r}\right)=\left\{ \begin{array}{l} 0,\;\frac{x^{2}+y^{2}}{a^{2}}+\frac{z^{2}}{c^{2}}\leq 1\\ \infty ,\;\frac{x^{2}+y^{2}}{a^{2}}+\frac{z^{2}}{c^{2}}>1 \end{array}\right.\;,\;a\ll c$$

Let us note that the problem in the frame of adiabatic approximation has been solved in [34]. On *z*-axis, we have “slow” motion; on *xy*-directions, we have “fast” motion.

The size quantization energy in the *XOY* plane is determined by the small semiaxis a and is of the order $\frac{\hbar^{2}}{\mu a^{2}}$. In turn, the energy of interaction between particles is of the order $\frac{1}{a}$. Therefore, in the *XOY* plane, size quantization energy is much stronger than the interaction one. So, the interparticle interaction in the radial direction can be considered weak in comparison with the size quantization, taking into account the small thickness of the system in this direction, and then, we consider the interaction between particles in axial direction as one-dimensional:(3)$$V_{\mathrm{int}}\left(1,\ldots ,N\right)=\sum_{i<j}\upsilon \left(\left|z_{i}-z_{j}\right|\right)$$

According to the adiabatic approach, in this case, the wave function of the system can be represented in the form
(4)$$\Phi \left({\vec{r}}_{1},\ldots ,{\vec{r}}_{N}\right)=\phi \left({\vec{\rho }}_{1}\left(z_{1}\right),\ldots ,{\vec{\rho }}_{N}\left(z_{N}\right)\right)\cdot \psi \left(z_{1},\ldots ,z_{N}\right)$$
where $\psi \left(z_{1},\ldots ,z_{N}\right)$ is the wavefunction describing the slow subsystem in the axial direction and
(5)$$\phi \left({\vec{\rho }}_{1}\left(z_{1}\right),{\vec{\rho }}_{2}\left(z_{2}\right),\ldots ,{\vec{\rho }}_{N}\left(z_{N}\right)\right)=\prod_{j=1}^{N}f_{j}\left({\vec{\rho }}_{j}\left(z_{j}\right)\right)$$

$f_{j}\left({\vec{\rho }}_{j}\left(z_{j}\right)\right)$ is a one-particle wavefunction in a 1D infinite quantum well with diameter
(6)$$d_{\rho }=2a\sqrt{1-\frac{z^{2}}{c^{2}}}$$
and this wavefunction has the following analytic form:(7)$$f\left(\vec{\rho }\left(z\right)\right)=C_{n_{\rho },\left|m\right|}e^{im\varphi }J_{m}\left(\kappa_{n_{\rho },m}\rho \left(z\right)\right)$$
where $n_{\rho },\;m$ are radial and magnetic quantum numbers, $J_{m}$ is Bessel functions of the first kind, $C_{n_{\rho },\left|m\right|}$ is normalization constant, $\kappa_{n_{\rho },m}=\sqrt{\frac{2\mu E_{n_{\rho }}}{\hbar^{2}}}$, and $E_{n_{\rho }}$ is energy of one-particle states in quantum well.

After substituting (7) in the Schrodinger equation, it can be shown that hole gas is localized in 1D parabolic potential [34]
(8)$$V_{conf}\left(z\right)=\frac{\mu \Omega^{2}z^{2}}{2}$$
with
(9)$$\Omega =\frac{1}{\sqrt{2}}\frac{\hbar \alpha_{1,0}}{\mu ac}$$

$\alpha_{1,0}$ is Bessel function zero of the first kind.

For the pair-interacting, N-particle system in this approximation, we have the following Schrodinger equation
(10)$$-\frac{\hbar^{2}}{2\mu }\sum_{i=1}^{N}\frac{\partial^{2}\psi }{\partial z_{i}^{2}}+\left\{\frac{\mu \Omega^{2}}{2}\sum_{i=1}^{N}z_{i}^{2}+\sum_{i<j}^{N}\upsilon \left(\left|z_{i}-z_{j}\right|\right)\right\}\psi =\left(E-NE_{0}\right)\psi =\varepsilon \left(N\right)\psi$$
where $E_{0}$ is the ground state of $E_{n_{\rho }}$.

## 3. Implementation of the One-Dimensional Moshinsky Model

Equation (10) is rather complicated, but there is a successful approximation for pair-interacting gas, which allows us to give an exact solution to equation (10). This approximation was proposed by M. Moshinsky to describe the behavior of nucleons in nuclei [37]. According to this model, particles interact with each other according to the oscillatory law and the interaction energy is proportional to the square of the distance between the particles.

In accordance with the foregoing, we rewrite the interaction potential in the following form
(11)$$V_{\mathrm{int}}\left(1,\ldots ,N\right)=\sum_{i<j}^{N}\gamma {\left(\left|z_{i}-z_{j}\right|\right)}^{2}$$
where γ is the interaction parameter, and for (10) we obtain
(12)$$-\frac{\hbar^{2}}{2\mu }\sum_{i=1}^{N}\frac{\partial^{2}\psi }{\partial z_{i}^{2}}+\left\{\frac{\mu \Omega^{2}}{2}\sum_{i=1}^{N}z_{i}^{2}+\sum_{i<j}^{N}\gamma {\left(\left|z_{i}-z_{j}\right|\right)}^{2}\right\}\psi =\varepsilon \left(N\right)\psi$$

We introduce the following notations
(13)$$\xi =\frac{z}{\sqrt{\frac{\hbar }{\mu \Omega }}},\;W\left(N\right)=\frac{E-NE_{0}}{\hbar \Omega },\;g=\frac{\gamma }{\mu \Omega^{2}}$$
and for the dimensionless Schrödinger equation, we get
(14)$$-\frac{1}{2}\sum_{i=1}^{N}\frac{\partial^{2}\psi }{\partial \xi_{i}^{2}}+\left\{\frac{1}{2}\sum_{i=1}^{N}\xi_{i}^{2}+g\sum_{i<j}^{N}{\left(\xi_{i}-\xi_{j}\right)}^{2}\right\}\psi =W\left(N\right)\psi$$

In order to solve the wavefunction of this system, we transform the original coordinates into Jacobi ones [40]:(15)$$\left\{ \begin{array}{l} Z=\frac{\xi_{1}+\ldots +\xi_{N}}{\sqrt{N}}\\ Z_{i}=\sqrt{\frac{i-1}{i}}\left(\xi_{i}-\frac{1}{i-1}\sum_{k=1}^{i-1}\xi_{k}\right),\;\;i=2,3,\ldots ,N \end{array}\right.$$

By using such transformations, we can separate (14) into two parts
(16)$$\left\{\left(-\frac{1}{2}\frac{\partial^{2}}{\partial Z^{2}}+\frac{Z^{2}}{2}\right)+\sum_{i=2}^{N}\left(-\frac{1}{2}\frac{\partial^{2}}{\partial Z_{i}^{2}}+\frac{1}{2}{\tilde{\Omega }}^{2}Z_{i}^{2}\right)\right\}\psi =W\left(N\right)\psi$$
where $\tilde{\Omega }=\sqrt{1+2Ng}$. The first part of (16) describes the center of mass motion and the second one describes the relative motion of the system. For the energy spectrum of the system, we have
(17)$$E_{n_{cm},\left\{n_{rel_{i}}\right\}}=NE_{0}+\hbar \Omega \left(n_{cm}+\frac{1}{2}\right)+\hbar \Omega \tilde{\Omega }\sum_{i=2}^{N}\left(n_{rel_{i}}+\frac{1}{2}\right)$$
where $\left\{n_{cm},n_{rel_{i}}\right\}$ are quantum numbers for center of mass and relative motion. Additionally, we can derive the exact wavefunction (coordinate part) of the system
(18)$$\psi \left(Z,Z_{i}\right)\;=\frac{1}{\sqrt{2^{n_{cm}}n_{cm}!}}{\left(\frac{1}{\pi }\right)}^{1/4}e^{-Z^{2}/2}H_{n_{cm}}\left(Z\right)\prod_{i=2}^{N}\frac{1}{\sqrt{2^{n_{rel_{i}}}n_{rel_{i}}!}}{\left(\frac{\tilde{\Omega }}{\pi }\right)}^{1/4}e^{-\frac{\tilde{\Omega }}{2}Z_{i}^{2}}H_{n_{rel_{i}}}\left(\sqrt{\tilde{\Omega }}Z_{i}\right)$$

Let us discuss the obtained results. As it was mentioned above, the case of the strongly prolate ellipsoidal QD made of Ge/Si have been considered. The material parameters which have been used in the calculations are the following: $\mu =0.394m_{0}$, where $m_{0}$ is the free electron mass, $\varepsilon_{r}=14.4$, and $a_{B}=4\;nm$ is the Bohr radius.

Interaction parameter γ was calculated by equating the Moshinsky model potential with the Coulomb one, and for $z_{ij}=\left|z_{i}-z_{j}\right|=2a_{B}$, interaction parameter $\gamma =0.4\;meV/a_{B}^{2}$.

In Figure 2, Figure 3, Figure 4, Figure 5 and Figure 6, the energy diagrams for relative motion for N=2,3,4 cases are presented. As can be seen from figures in the pair-interacting hole gas case, a change in the geometric parameters of QDs has a tiny effect on the energy states of the relative motion. In the non-interacting gas case, the relative motion energy reacts very sharply to the changes in the geometric parameters of the QD. Moreover, the growth of the semi-axes of the ellipsoidal QD leads to the decrease in the inter-level distance (Figure 7).

In Figure 5, the energy diagrams for center of mass motion are presented. The energy and inter-level distance do not depend on N and γ. The growth of the ellipsoidal QD semi-axes leads to the decrease in the inter-level distance as in the relative motion case (Figure 7).

In Figure 6, the energy diagrams for a three-particle system are presented. As can be see from the figure, the growth of the interaction parameter γ leads to zone structure formation because inter-level distance does not change with γ increase for center of mass motion and increases for the relative one (Figure 8).

Figure 8 and Figure 9 present the dependences of the energies of hole gas on the interaction and number of particles. As can be seen from them, the motion of the mass center does not depend on the number of particles and interaction, and inter-level distance is not being changed with the increase of N and γ. In the case of relative motion with N increase, the interaction leads to the increase in energy and inter-level distance, which confirms Kohn’s theorem.

## 4. Kohn Theorem Realization

The presence of parabolic confinement potential in the axial direction leads hole gas Hamiltonian to the form (10). This form of the Hamiltonian corresponds to the one-dimensional, pair-interacting hole gas localized in a parabolic quantum well. The situation is similar to the case of electron gas localized in a parabolic quantum well investigated in [40]. In such a system, the conditions for the generalized Kohn’s theorem are realized. The specific geometry of the studied QD gives the conditions necessary for realization of the generalized Kohn’s theorem. In this point of view, we can note that in [36], the fulfillment of Kohn’s theorem for hole gas in a lens-shape Ge/Si QD was shown experimentally as well as theoretically. The necessary criterion for the fulfillment of this theorem is the pair-interaction model, which depends only on the interparticle distance. This condition is fully satisfied for a hole gas with interaction model (11). Let us demonstrate the above.

Let, now, a long-wavelength electromagnetic perturbation with the electric component $E\left(t\right)=E_{0}e^{-i\omega t}$ fall on the system. In the mentioned approximation, the perturbation operator is written as
(19)$$H_{1}=-e\sum_{i=1}^{N}z_{i}E\left(t\right)$$

Now we introduce the creation and annihilation operators in the following form
(20)$${\hat{C}}_{z}^{\pm }={\left(\frac{\mu \Omega }{2\hbar }\right)}^{1/2}\sum_{i=1}^{N}\left(z_{i}\mp \frac{i}{\mu \Omega }{\hat{p}}_{z_{i}}\right)$$
via ${\hat{C}}_{z}^{\pm }$ we rewrite Hamiltonian of the system in the following form
(21)$$\hat{H}={\hat{H}}_{0}+\sum_{i<j}^{N}\gamma {\left(z_{i}-z_{j}\right)}^{2}$$
where
(22)$${\hat{H}}_{0}=\sum_{i=1}^{N}\left(-\frac{\hbar^{2}}{2\mu }\sum_{i=1}^{N}\frac{\partial^{2}}{\partial z_{i}^{2}}+\frac{\mu \Omega^{2}}{2}\sum_{i=1}^{N}z_{i}^{2}\right)=\hbar \Omega \left({\hat{C}}_{z}^{+}{\hat{C}}_{z}^{-}+\frac{1}{2}\right)$$
then one can check by a direct calculation that the following commutation relations take place
(23)$$[{\hat{H}}_{0},{\hat{C}}_{z}^{\pm }]=\pm \hbar \Omega {\hat{C}}_{z}^{\pm }$$

If we consider the following commutator
(24)$$\left[{\hat{C}}_{z}^{+},\sum_{i<j}^{N}\gamma {\left(z_{i}-z_{j}\right)}^{2}\right]=\left[{\left(\frac{\mu \Omega }{2\hbar }\right)}^{1/2}\sum_{i=1}^{N}\left(z_{i}-\frac{i}{\mu \Omega }{\hat{p}}_{z_{i}}\right),\sum_{i<j}^{N}\gamma {\left(z_{i}-z_{j}\right)}^{2}\right]$$
then we arrive at the following commutators
(25)$$\left\{ \begin{array}{l} K_{1}=\left[{\left(\frac{\mu \Omega }{2\hbar }\right)}^{1/2}\sum_{i=1}^{N}z_{i},\gamma \sum_{i<j}^{N}{\left(z_{i}-z_{j}\right)}^{2}\right],\\ K_{2}=\left[-\frac{i}{\sqrt{2\mu \hbar \Omega }}\sum_{i=1}^{N}{\hat{p}}_{z_{i}},\gamma \sum_{i<j}^{N}{\left(z_{i}-z_{j}\right)}^{2}\right]. \end{array}\right.$$

From (25), it is clear that $K_{1}=K_{2}=0$, so commutator (24) is equal to 0 so
(26)$$[\hat{H},{\hat{C}}_{z}^{\pm }]=\pm \hbar \Omega {\hat{C}}_{z}^{\pm }$$

If the function $\psi_{\left\{n\right\}}^{0}$ is the eigenfunction of the operator ${\hat{H}}_{0}$ with the eigenvalues $E_{\left\{n\right\}}^{0}$, then, respectively, function $C_{z}^{+}\psi_{\left\{n\right\}}^{0}$ is the eigenfunction of the operator ${\hat{H}}_{0}$ but with the energy $E_{\left\{n\right\}}^{0}+\hbar \Omega$. A similar conclusion holds for $\hat{H}$. Thus, both in the case of hole gas with allowance for the interparticle interaction and in the case when this interaction is absent, under the action of a long-wavelength radiation, the dipole transitions, determined by the frequency Ω, take place, which is the essence of Kohn’s theorem (Figure 8 and Figure 9).

## 5. Conclusions

In the present article, the realization of the Kohn theorem has been shown for the prolate ellipsoidal Ge/Si quantum dot. The long-wave absorption is considered by the few-hole gas. The interparticle interaction is considered only in axial direction and the Moshinsky approximation has been used for the quantitative description. According to this model, particles interact with each other according to the oscillatory law and the interaction energy is proportional to the square of the distance between the particles. In the pair-interacting hole gas case, a change in the geometric parameters of QDs has a tiny effect on the energy states of the relative motion, while in the non-interacting gas case, the relative motion energy reacts very sharply to the changes in the geometric parameters of the QD. The motion of the center of mass does not depend on the number of particles and interaction, and inter-level distance is not being changed with the increase of N and γ. In the case of relative motion with N increase, the interaction leads to the increase in energy and inter-level distance, which confirms Kohn’s theorem.

## Figures and Tables

**Figure 1 nanomaterials-10-01896-f001:**
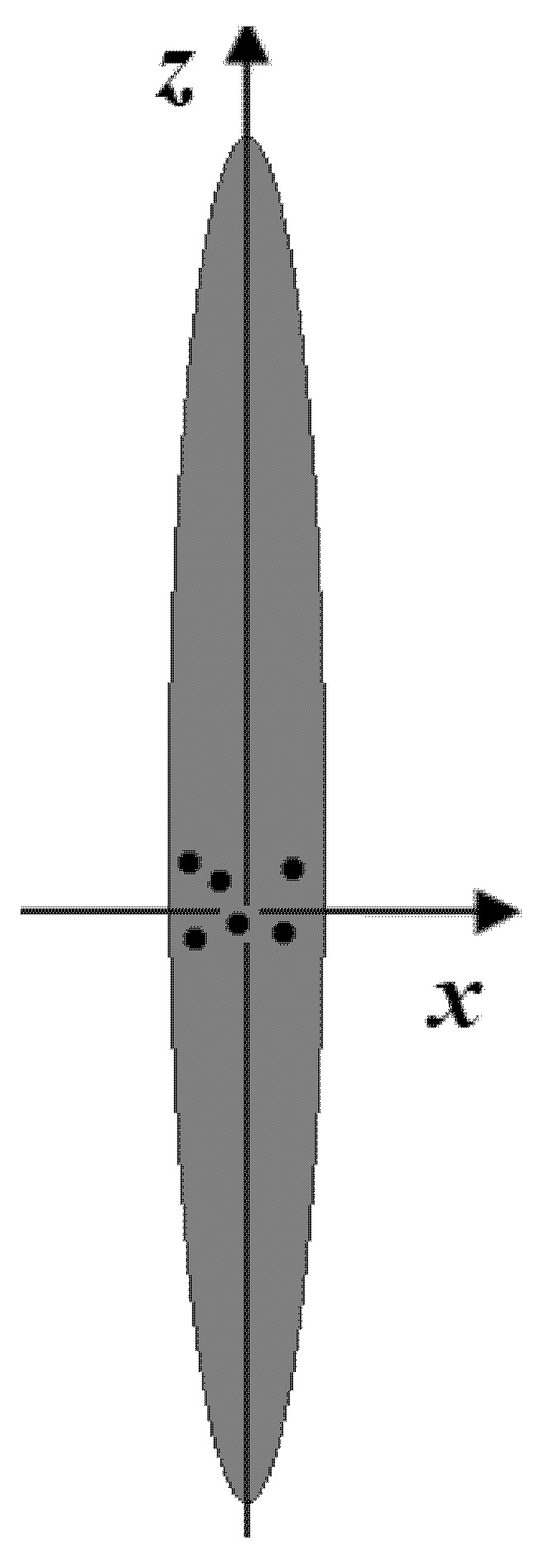
Cross-section of the strongly prolate ellipsoidal quantum dot (QD) with few-hole gas.

**Figure 2 nanomaterials-10-01896-f002:**
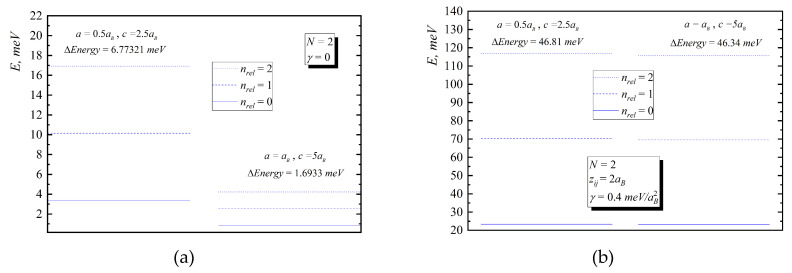
Energy diagram ((**a**)—without interaction, (**b**)—with interaction) for relative motion for a two-particle case for the following values: I. $a=0.5a_{B},c=2.5a_{B}$, II. $a=a_{B},c=5a_{B}$.

**Figure 3 nanomaterials-10-01896-f003:**
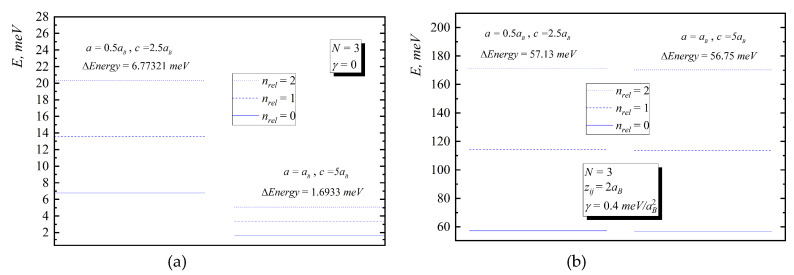
Energy diagram ((**a**)—without interaction, (**b**)—with interaction) for relative motion for three-particle case for the following values: I. $a=0.5a_{B},c=2.5a_{B}$, II. $a=a_{B},c=5a_{B}$.

**Figure 4 nanomaterials-10-01896-f004:**
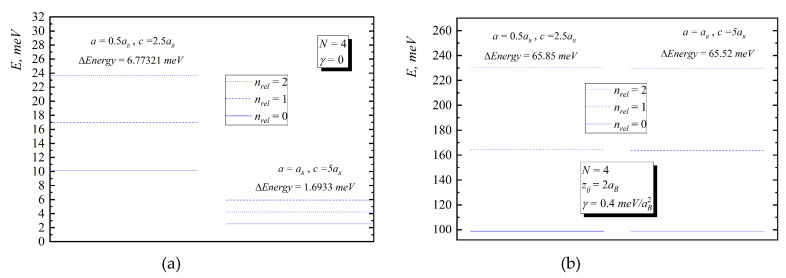
Energy diagram ((**a**)—without interaction, (**b**)—with interaction) for relative motion for four-particle case for the following values: I. $a=0.5a_{B},c=2.5a_{B}$, II. $a=a_{B},c=5a_{B}$.

**Figure 5 nanomaterials-10-01896-f005:**
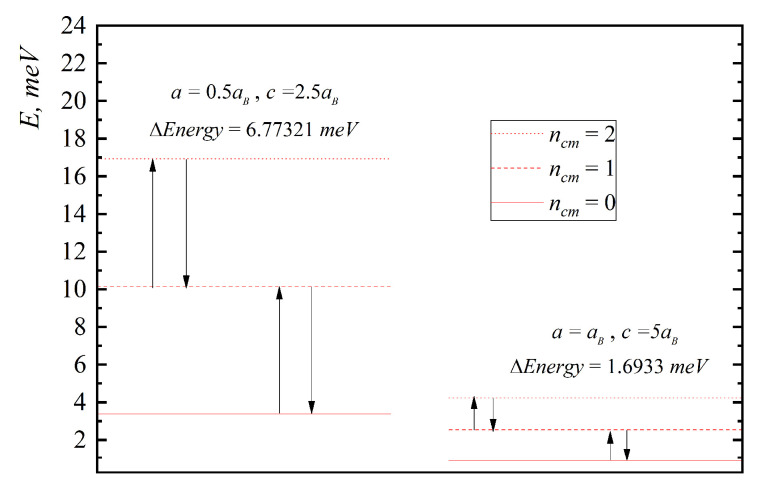
Energy diagram for the center of mass motion for the following values: I. $a=0.5a_{B},c=2.5a_{B}$, II. $a=a_{B},c=5a_{B}$.

**Figure 6 nanomaterials-10-01896-f006:**
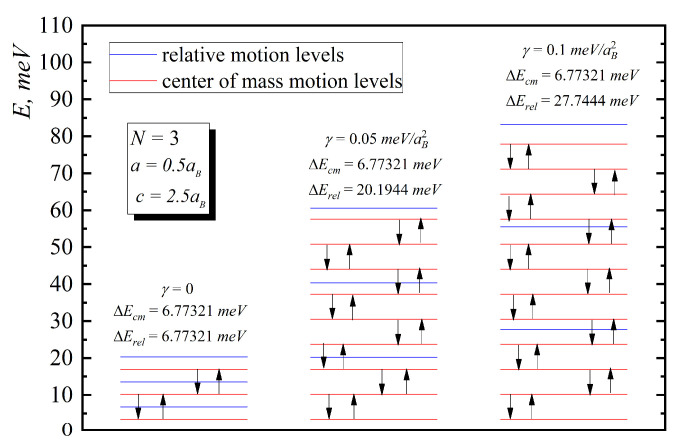
Energy diagram for three-particle system for the following values: I. γ=0, II. $\gamma =0.05\;meV/a_{B}^{2}$, III. $\gamma =0.1\;meV/a_{B}^{2}$.

**Figure 7 nanomaterials-10-01896-f007:**
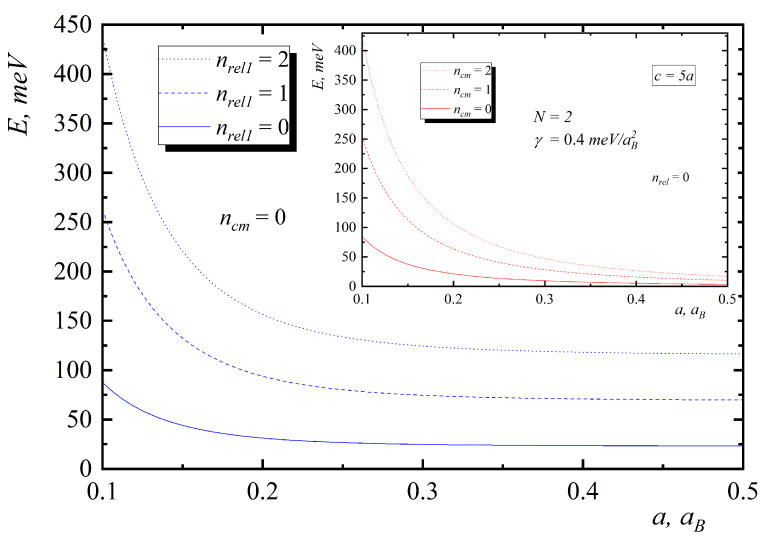
Dependences of the energies on the small semiaxis for relative motion (insert—center of mass motion).

**Figure 8 nanomaterials-10-01896-f008:**
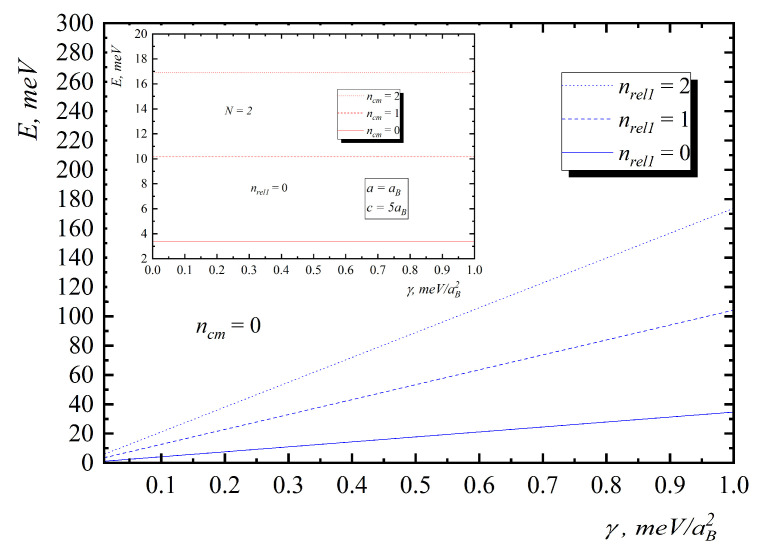
Dependences of the energies on the interaction parameter γ for relative motion (insert—center of mass motion).

**Figure 9 nanomaterials-10-01896-f009:**
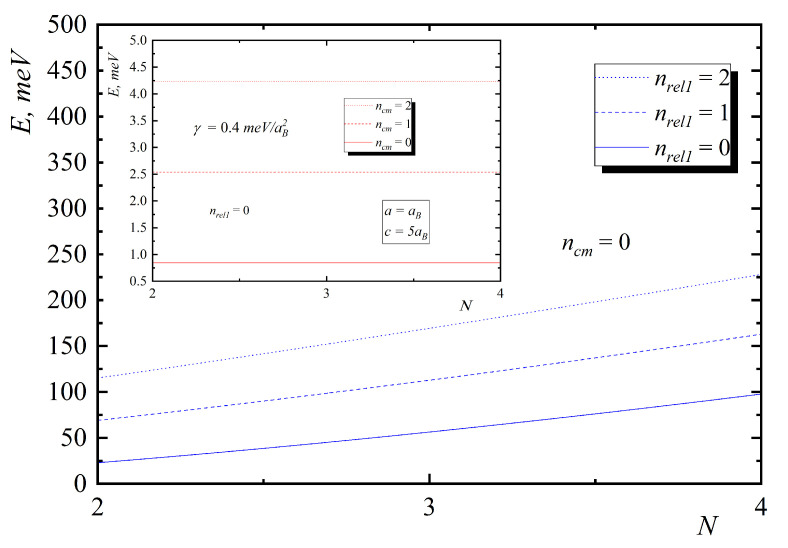
Dependences of the energies on the number of particles for relative motion (insert – center of mass motion).

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
