# Peer review of "Long-wave Absorption of Few-Hole Gas in Prolate Ellipsoidal Ge/Si Quantum Dot: Implementation of Analytically Solvable Moshinsky Model"

_nanomaterials, 2020, doi:10.3390/nano10101896_

Round 1

Reviewer 1 Report

The authors present an analytically tractable model for few-hole states in elongated quantum dots. By separating the transverse confinement and the motion in z axis, they motivate the application of the Moshinski model, which uses a harmonic interactions instead of the more realistic Coulomb interaction. The Moshinski model was originally introduced as a somewhat artificial, but  analytically solvable model for the Hartree-Fock states of an interacting fermion system. The authors then include a dipole interaction term to discuss  the interaction of long-wavelength electromagnetic fields the holes of their model system.

The approach of the authors strikes me as being somewhat mathematical and using/constructing the physical system more as a realization of their solvable model.  In the framework of their model, their derivations and results are well described and can be easily understood by the reader. I was confused by their modeling of the interaction with electromagnetic fields: I would have expected an electron-hole transition instead of a dipole interaction constructed from the z-component of the holes. The authors need to comment on this.

I recommend publication, but I would like to caution the authors that the interest in these systems in the community seems to have largely died down in the last 20 years. I do not see how an experimentalist could get anything out of their results, or whether they are making any methodological progress. I would urge the authors to keep these questions in mind when the pick their research problems.

Author Response

Dear Editor

I am sending you the revised version of the manuscript entitled “Long-wave absorption of few-hole gas in prolate ellipsoidal Ge/Si quantum dot: implementation of analytically solvable Moshinsky model” taking into consideration the reviewers’ comments and recommendations. The authors are grateful to the reviewers for careful and detailed reading of the manuscript and useful remarks and recommendations. The manuscript has been revised taking into account all suggestions of the reviewers. In revised version of the manuscript the added parts are highlighted by yellow.

Comments for Reviewer 1

Motivation: We agree with reviewer that there is currently no great interest of the scientific community in the generalized Kohn theorem. However, the main motivation for considering this problem was the experimental results obtained in 2016 by the group of Professor Dmitry Firsov from St. Petersburg Polytechnic University on the study of long-wave absorption in strongly oblate lens-shape Ge/Si quantum dots (QDs). The specific geometry of QDs and the peculiarities of the band structure of heavy holes in Ge/Si QDs made it possible to experimentally observe this effect in a hole system. Since in the experiment the incident radiation is long-wave, then it is quite reasonable to use the dipole approximation for perturbation operator. The dipole approximation is the key element for the Kohn theorem. Earlier, theoretically, the possibility of realizing the generalized Kohn's theorem in ellipsoidal and lens-shape QDs containing the electron gas was predicted by our group. It is noteworthy that the theoretical proof of the realization of this theorem is the same for both the hole and electron gases. These experimental and theoretical results were published in Nanomaterials where it was shown that the experimental results are in good coincidence with theory.

In the next step, we tried to consider a gas model, which can give, on the one hand, an analytical description of hole gas, and, on the other hand, have such character of interparticle interaction that will allow the theoretical realization of the generalized Kohn's theorem. This will make it possible to give a complete analysis of the energy spectrum of the system, both the motion of the center of mass and the relative one.

Model selection: We agree with the reviewer that the choice of the Moshinsky model is due to its exact solvability; this is a successful mathematical model. It is clear that the real potential is the Coulomb one. However, for the long-wave transitions between the center of mass (which is the content of Kohn's theorem), the form of interparticle interaction potential isn’t important (the main is pair-interacting type of the potential and decency only from the modulus of the mutual distance between the particles). On the other hand, the Moshinsky model allows to construct the exact energy spectrum of few-particle system and makes it possible to see a clear picture of quantum transitions, in particular, between the center of mass levels (it simplifies the understanding of Kohn's theorem; the picture of quantum transitions is very clear). The purpose was to build a bridge between the nuclear physics model and the QD model containing a pair-interacting particles gas. Since the one-dimensional model of Moshinsky is relatively simple, we decided to consider the geometry of strongly prolate ellipsoidal QD.

Reviewer 2 Report

The present paper is an analytical study of a 1D hole-gas in an artificial atom in a parabolic potential. The implementation the authors refer to, is a prolate ellipsoidal Ge/Si quantum dot.
If the major axis is much bigger than the minor, it is shown that the relative coordinates and the center of mass can be separated in a adiabatic framework, then the system is mapped into the exact solvable Maoshinsky Model. A spectral analysis for a few-hole system is done and then the validity of the Generalized Kohn theorem is proven.

Ge/Si quantum dots represents very interesting and promising artificial systems for a lot of applications and this analysis can be of interest in the analysis of the spectral properties of such systems. Moreover the validity of the Kohn theorem could be very important for possible interfaces with photons.

The paper is written in a clear form, is original and scientifically rigorous.
I think that the analysis is relevant in the context of Si/Ge quantum dots.
For this reason in my opinion the paper is suitable to be published on Nanomaterials, after considering the following comments:

1. The authors introduce the Kohn theorem referring to refs [4,26,27]. The citations are appropriate, but I think that the theorem should be at least enunciated and discussed a little bit also in the paper.

2. In Fig.1 the major axis of the ellipse is along the y-direction while in the text it is along the z direction.

3. I suggest to revise the Sec. 2, where the adiabatic approximation is discussed, in order to make it self consistent and more readable to a wide audience. I understand that great part of this section is standard calculation, but I find that some steps are missing and some more comments are worth. For example:
a. Is is written that "The interparticle interaction in radial direction can be considered weak in comparison with the size quantization taking into account the small thickness of the system..". I think that this approximation should be discussed more.
b. In formula (4) neither the \psi(z_1,..,z_N) function nor \vec{\rho) are defined
c. In formula (5) I think that a j index should be added to f.

4. In general, I think that a discussion on the motivations to analyze the validity of the Kohn theorem is missing. I thing the authors should add some comments on the possible implications of this result. Moreover, could the occurrence of Kohn theorem be os some importance in the study os dissipation and decoherence effects due to interactions with phonons?

Author Response

Dear Editor

I am sending you the revised version of the manuscript entitled “Long-wave absorption of few-hole gas in prolate ellipsoidal Ge/Si quantum dot: implementation of analytically solvable Moshinsky model” taking into consideration the reviewers’ comments and recommendations. The authors are grateful to the reviewers for careful and detailed reading of the manuscript and useful remarks and recommendations. The manuscript has been revised taking into account all suggestions of the reviewers. In revised version of the manuscript the added parts are highlighted by yellow.

Comments for Reviewer 2

  1. The authors introduce the Kohn theorem referring to refs [4,26,27]. The citations are appropriate, but I think that the theorem should be at least enunciated and discussed a little bit also in the paper.

We have added some part with Kohn theorem discussing in introduction:

This theorem was originally formulated for the case of electron gas in axial magnetic field by Walter Kohn in 1961. Kohn had shown that when in dipole approximation a long-wavelength electromagnetic perturbation fell on the short-range pair-interacting electron gas with the interparticle interaction energy with the following form, the frequency of the cyclotron resonance was not affected by the number of electrons. Maxim and Chakraborty generalized this theorem for the case of QDs [4]. In this paper authors have considered a pair-interacting electron gas localized in a two-dimensional symmetric parabolic QD, in the presence of an axial magnetic field.

  1. In Fig.1 the major axis of the ellipse is along the y-direction while in the text it is along the z direction.

We have corrected this technical mistake.

  1. I suggest to revise the Sec. 2, where the adiabatic approximation is discussed, in order to make itself consistent and more readable to a wide audience. I understand that great part of this section is standard calculation, but I find that some steps are missing and some more comments are worth. For example:
  2. It is written that "The interparticle interaction in radial direction can be considered weak in comparison with the size quantization taking into account the small thickness of the system.". I think that this approximation should be discussed more.

We have added discussing part about this question after fig.1:

The size quantization energy in the XOY plane is determined by the small semiaxis  and is of the order . In turn, the energy of interaction between particles is of the order . Therefore, in the XOY plane, size quantization energy is much stronger than interaction one. So, the interparticle interaction in radial direction can be considered weak in comparison with the size quantization taking into account the small thickness of the system in this direction, and then, we will consider the interaction between particles in axial direction as one dimensional:

  1. In formula (4) neither the \psi(z_1,..,z_N) function nor \vec{\rho) are defined

We have added the explanation of the axial wavefunction (see 122 line):

where  is wavefunction describing slow subsystem in axial direction and
c. In formula (5) I think that a j index should be added to f.

We have added j index to f.

  1. In general, I think that a discussion on the motivations to analyze the validity of the Kohn theorem is missing. I thing the authors should add some comments on the possible implications of this result. Moreover, could the occurrence of Kohn theorem be as some importance in the study of dissipation and decoherence effects due to interactions with phonons?

We have added our motivation to investigate the long-wave transitions in this system and possible application of discussed results:

The presence of a parabolic confinement potential in the axial direction leads hole gas Hamiltonian to the form (10). This form of the Hamiltonian corresponds to a one-dimensional pair-interacting hole gas localized in parabolic quantum well. The situation is similar to the case of electron gas localized in parabolic quantum well investigated in [40]. In such system, the conditions for the generalized Kohn's theorem are realized. The specific geometry of the studied QD gives the conditions necessary for realization of the generalized Kohn's theorem. In this point of view, we can note, that in [36] the fulfillment of Kohn's theorem for hole gas in lens-shape Ge/Si QD was shown experimentally as well as theoretically. Since the necessary criterion for the fulfillment of this theorem is the pair-interaction model, which depends only of the interparticle distance. This condition is fully satisfied for a hole gas with interaction model (11).

In our opinion, in the future one of the possible applications of Kohn theorem in the field of semiconductor optoelectronic devices can be construction of photon amplifier. Regarding the question could the occurrence of Kohn theorem be as some importance in the study of dissipation and decoherence effects due to interactions with phonons, we should mention that we are not competence in such topics and we cannot give an exhaustive answer to this question. On the other hand, we think that the problem mentioned by the reviewer is actual, timely and very important.
